# Bearing Fault Diagnostics Based on the Square of the Amplitude Gains Method

**Rafał Grądzki** [1,*] 🆔 **, Błażej Bartoszewicz** [1] **and José Emiliano Martínez** [2]

[1] Faculty of Mechanical Engineering, Bialystok University of Technology, Wiejska st. 45C, 15-351 Bialystok, Poland

[2] Department of Engineering Studies for Innovation, Universidad Iberoamericana Ciudad de México, Prol. Paseo de la Reforma 880, CDMX 01219, Mexico

* Correspondence: r.gradzki@pb.edu.pl

**Abstract:** The article presents an adaptation of a parametric diagnostic method based on the square of the amplitude gains model, which was tested in experimental studies on bearing damage detection (outer race, inner race, bearing balls damage). The described method is based on the shaft displacement signal analysis, which is affected by vibrations coming from the bearings. The diagnostic model's parameters are determined by processing the signal from the time domain to the frequency domain in a few steps. Firstly, the recorded signal is divided into two observation periods, next the analytical autocorrelation functions are determined and approximated by a polynomial. Then, the diagnostic thresholds are adopted, and the model parameters are converted into damage maps that are easy to interpret and assess the technical condition of the bearings. The presented method shows the technical condition of bearings in a qualitative way. Depending on the received color damage maps, it is possible to determine their level of wear. *Green* and *blue* indicate poor wear or no damage, *red* indicates increased wear, and *black* clearly indicates a damaged bearing.

**Keywords:** bearing; bearing outer race; bearing inner race; diagnostic model; diagnostics; bearing damage; the square of the amplitude gain

## 1. Introduction

Bearing failure diagnostics is an important part of rotating machinery maintenance. Accurate and early bearing damage detection contributes to safer and more efficient machine exploitation [1–12].

Following many hours of bearing operation, the impact of excessive vibrations in combination with limited lubrication of machine parts leads to bearing damage: outer race, inner race, damage to the cage, and rolling elements (balls). If left undetected for some time, this mechanical damage can cause equipment failure and, consequently, unscheduled downtime. Therefore, timely intervention or preventive maintenance is vital to keeping rotating equipment running efficiently.

Recently, there have been many techniques that can be used to monitor bearing health, such as vibration monitoring [13–29], noise monitoring [30–32], temperature monitoring [33–35], and residual wear analysis [36–39]. However, vibration monitoring is the most effective technique—single-point defects produce successive pulses with each contact of the damage with the rolling element, and each contact can excite a high-frequency resonance in the overall structure. The mentioned analysis allows for detecting, locating, and distinguishing various types of damage from the moment of its occurrence before they become critical and dangerous. These damages can be dispersed or localized [22].

Unfortunately, the vibration signal does not only contain signals originating directly from the bearing. It also contains vibration signals from cooperating elements, e.g., shafts or another mechanism. Their misalignment, unbalance, stiffness, clearance, and friction will also affect the recorded signal coming from the bearings. Therefore, modern diagnostics

require techniques of advanced signal processing: fast Fourier transform (FFT) [40–50], cepstrum analysis (CA) [51–53], short time Fourier transform (STFT) [54,55], Wigner–Ville Distribution (WVD) [54–56], the envelope analysis (EA) [57,58], and wavelet transform (WT) [58–62], and various advanced models based on artificial intelligence [63–80].

Vibration data analysis mainly includes the time domain, frequency domain, time-frequency domain, and other analysis methods, but the vast majority are based on time-frequency analysis.

The authors also developed time-frequency methods. In their work, they proved the effectiveness of parametric methods: in the form of a square signal amplitude amplification and an original model in the form of a difference in phase shifts of signals for diagnosing shafts and compressors of aircraft engine turbines.

A mathematical model of a square of amplitude gains for diagnostic purposes was first presented in 2004 [81]. Lindstedt and Kotowski presented its application for detecting damage to an aircraft engine blade in stationary conditions (removal of the blade from the engine required). The vibration signal was generated by modal hammer impact. In 2007, Kotowski and Lindstedt [82] presented a complete study with an analysis of the impact of this model's parameters on the damage type and location.

The next stage of the method development took place at the Air Force Institute of Technology (ITWL) in Warsaw (Poland) in 2009 [83,84]. The article presents the mathematical basis for adapting the method to diagnostics during a rotating machine operation. However, this approach has not been experimentally verified. In 2010, Lindstedt and Grądzki [85] determined the parameters of the diagnostic model from the recorded signal from the inductive sensor. In 2012, Grądzki [86] in his doctoral thesis, presented extensive research on the compressor blade of the S0–3 turbine engine during its operation. An analysis of the impact of changes in the environment represented by the rotational speed on the changes of model parameters was carried out, and color damage maps were used to analyze the technical condition. For verification, a new diagnostic model in the form of a difference in phase shifts of signals was also developed and tested. Endoscopic measurements additionally verified all results.

In 2018, the authors [87], using a parametric diagnostic model of a difference in phase shifts of signals, presented an analysis of the technical condition of the entire rim of SO-3 compressor blades. In 2020 [88], they presented another model modification, which allowed both to examine the technical condition of the blade and its stationary condition during operation.

The model of a square of amplitude gains has also been implemented for shaft diagnostics during their operation [89]. The authors showed the effectiveness of the damage map to verify the different types of shaft damage in simulation and experimental research.

In the presented article, the authors adapted the square of amplitude gains method to detect bearing damage (outer race, inner race, bearing ball damage at different depths). Firstly, a recorded signal of shaft displacement is divided into two observation periods. Then, the analytical autocorrelation functions are determined and approximated by a polynomial. On their basis, the Fourier transform is used to convert to the spectral (frequency) form. Thanks to this, it is possible to determine the parameters of the model to which fixed statistical diagnostic thresholds are assigned. The last step is to convert the model parameters into color damage maps that are easy to analyze and interpret.

At this stage, the proposed method allows for early detection of bearing technical conditions (damaged or undamaged). However, based on the parameter values, the authors cannot indicate the type of damage.

The article is presented as follows: The introduction is presented in Section 1. Section 2 describes the mathematical procedure for determining the model's parameters and obtaining color damage maps of the bearings. Section 3 presents the measurement stand and describes the objective and scope of the tests. Section 4 presents the experimental results of the tests carried out and compares the results obtained (by using FFT analysis and a square

of the amplitude gains method) for the undamaged and damaged bearings. Conclusions and discussions are in Section 5.

## 2. Condition Monitoring Based on the Squared Amplitude Gain of Signals

The mathematical foundations of the method are described in detail in [89].

To use the proposed diagnostic model to monitor the condition of the bearings, signals of rotor displacements near the supporting bearings during machine operation are measured and sampled. It is assumed that the rotor is undamaged, its state remains unchanged, and various bearing damage variants are introduced simultaneously. In this way, researchers determine the condition of the bearings, using the method described in the article based on the rotor displacement signals. Finally, as a result of using the method, damage maps characterizing the technical condition of the bearings are obtained.

In the first step of analyzing the signals obtained from the experimental studies, the time interval $T_{02}$, hereinafter referred to as a cycle, is determined for every single revolution. This interval corresponds to a full revolution of the shaft reduced on the outer parts of the intervals by several samples so that successive cycles do not overlap. Then the interval $T_{02}$ is divided into two separate time intervals $T_{01}$ and $T_{12}$ (sample sets). Point $T_1$ is common for both ranges and marks the moment when the shaft surface is farthest from the sensor face (Figure 1).

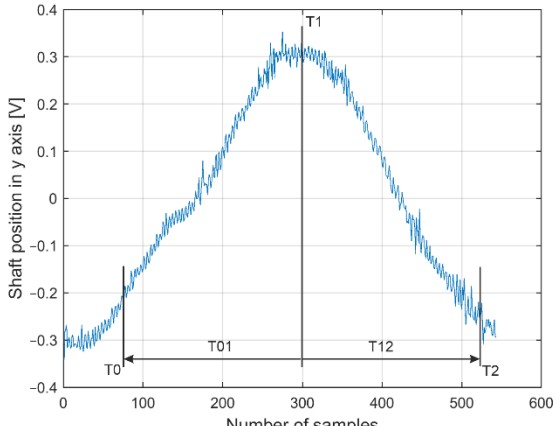

**Figure 1.** Dividing the signal into two sub-periods $T_{01}$ and $T_{12}$.

Each cycle's number of signal samples must be identical and selected to ensure the proper statistical evaluation of the measured operational signals. For the tested rotational speed, the intervals $T_{01}$ and $T_{12}$ contain the same number of signal samples, and these intervals do not overlap. Frequency leakage was reduced by scaling each interval by a Hanning window.

For the displacement signal $y(t)$ in the assumed time intervals $T_{01}$ and $T_{12}$, estimates of the autocorrelation functions $R_{yy}{}^{T01}$ and $R_{yy}{}^{T12}$, are determined, which are then approximated by analytical expressions (polynomials), ensuring a fit above 0.997.

The correlation function takes the form:

$$R_{yy}(\tau) = a_n \tau^n + \ldots + a_4 \tau^4 + a_3 \tau^3 + a_2 \tau^2 + a_1 \tau + a_0 \tag{1}$$

where: $a_0, a_1, \ldots a_n$—coefficients of the polynomials, $n$ = 0, 1, 2, 3, $\ldots$, $r$.

The order $n$ of the polynomials should be selected carefully—a too low order will result in inaccurate approximations, a too high order will result in an excessive number of polynomial coefficients and longer calculation times.

The described parametric diagnostic model is based on the functions of the spectral density of the power of the registered rotor vibration signal $y(t)$ in two time intervals ($T_{01}$, $T_{12}$) and the environment $x(t)$. It is assumed that the rotor work signals $y(t)$ and the envi-

ronment $x(t)$ are stochastic and the disturbed time courses are expressed by autocorrelation functions $R_{xx}(\tau)$ and $R_{yy}(\tau)$.

Based on the analytical forms of the auto-correlation function, the corresponding power auto-spectral density functions $S_{yy}^{T01}(j\omega)$ and $S_{yy}^{T12}(j\omega)$ are determined using the Fourier transform:

$$S_{yy}^{T01}(j\omega) = F(R_{yy}^{T01}(\tau)) \tag{2}$$

$$S_{yy}^{T12}(j\omega) = F(R_{yy}^{T12}(\tau)) \tag{3}$$

In the next step, it is assumed that the power densities of the environment signal $x(t)$ in time $T_{01}$, $T_{12}-S_{xx}^{T01}(j\omega)$, and $S_{xx}^{T12}(j\omega)$ have been determined (in the same way as $S_{yy}$). Since the observation times are very close to each other, it can be assumed that the environment has not changed at that time; therefore $S_{xx}^{T01}(j\omega) \cong S_{xx}^{T12}(j\omega)$. Based on the above condition, a parametric diagnostic model can be determined in the form of the quotient of the power density function (square of the amplitude gains), allowing to eliminate the environment:

$$A_{T12T01}^2(\omega) = \frac{\frac{S_{yy}^{T12}}{S_{xx}^{T12}}}{\frac{S_{yy}^{T01}}{S_{xx}^{T01}}} \xrightarrow{S_{xx}^{T12} \cong S_{xx}^{T01}} \frac{S_{yy}^{T12}}{S_{yy}^{T01}} = \frac{A_0^* + A_1^*s + A_2^*s^2 + \ldots + A_n^*s^n}{B_0^* + B_1^*s + B_2^*s^2 + \ldots + B_n^*s^n} \tag{4}$$

where: $s$—complex variable, $s = j\omega$; $A_i^*$, $i = 0, 1, \ldots, n$—numerator estimates parameters; $B_i^*$, $i = 0, 1, \ldots, n$—denominator estimates parameters; $n$—polynomial order.

Despite the elimination of the environment, the parameters of the $A_{T12T01}^2$ model are directly related to the change in the technical condition of the rotor system supported on bearings. Therefore, a characteristic feature of the $A_{T12T01}^2$ model is that it does not require the measurement of environmental signals. However, it is indirectly considered by conducting diagnostic tests (two observation periods, determining the diagnostic model as a quotient of diagnostic models combining diagnostic signals and any environment with technical condition parameters).

Parameters of the numerator ($A_i$) and denominator ($B_i$) of the model are determined for each cycle of the rotating shaft, thus creating a matrix of parameters, describing the technical condition of the bearings during the operation-a damage map. The mean value $\mu$ and the standard deviation $\sigma$ are determined for the corresponding model parameters from each cycle (e.g., $A_0$ for each cycle—the first column in the matrix of parameters). On their basis, statistical diagnostic thresholds of the forms $\mu \pm \sigma$, $\mu \pm 2\sigma$, $\mu \pm 3\sigma$ are determined.

Then, the determined values of Ai and Bi parameters are compared to the determined diagnostic thresholds and changed to the appropriate color (Table 1), creating the so-called damage maps of the technical condition of the object:

- *green* color if the parameter value did not exceed $\mu \pm \sigma$,
- *blue* color if the parameter value exceeded $\mu \pm \sigma$,
- *red* color if the parameter value exceeded $\mu \pm 2\sigma$,
- *black* color if the parameter value exceeded $\mu \pm 3\sigma$.

**Table 1.** Legend for the damage maps.

| The Predominant Color in the Damage Map | Technical Condition | Stationary Condition |
|---|---|---|
| *Black* and *red* | "Serious failure" | "Strong" changes of bearing technical condition |
| *Red* | "Excessive wear" | "Weak" changes of bearing technical condition |
| *Green* and *blue* | "Slight or no wear" | "Slight" changes of bearing technical condition |

This way, a damage map of the bearing is obtained, showing its technical condition. For example, if there are many *black* colors, the bearing is damaged, *red* means increased bearing wear, and *green* and *blue* colors mean little or no wear. This approach shows an unambiguous picture of the bearing damage assessment.

## 3. Experimental Test Stand

The proposed damage detection method was experimentally verified on the stand operated at the Bialystok University of Technology (Figure 2 and Table 2). The main element of the test stand is the rotor mounted on two ball bearings (1a–on the brake side and 1b–on the drive side), driven by an electric motor (2) with adjustable speed (up to 2000 rpm). The rotor consists of three parts: the middle part is a replaceable shaft (3), and the outer parts are the drive shaft and the braking shaft supported on the bearings. The shafts are connected into one rotor using conical fits and flanges with bolts ensuring, on the one hand, the axial symmetry of the rotor and, on the other, quick and simple reconfiguration of the shafts. Mass discs (4) are attached to the outer shafts. At the shaft's end, an electromagnetic brake (7) enables the torsional load introduction. The radial positions of the shaft near both bearings are measured by eddy current sensors placed in the horizontal and vertical planes, two at each bearing. Variants of the bearing damage were implemented in the support closer to the drive (1b).

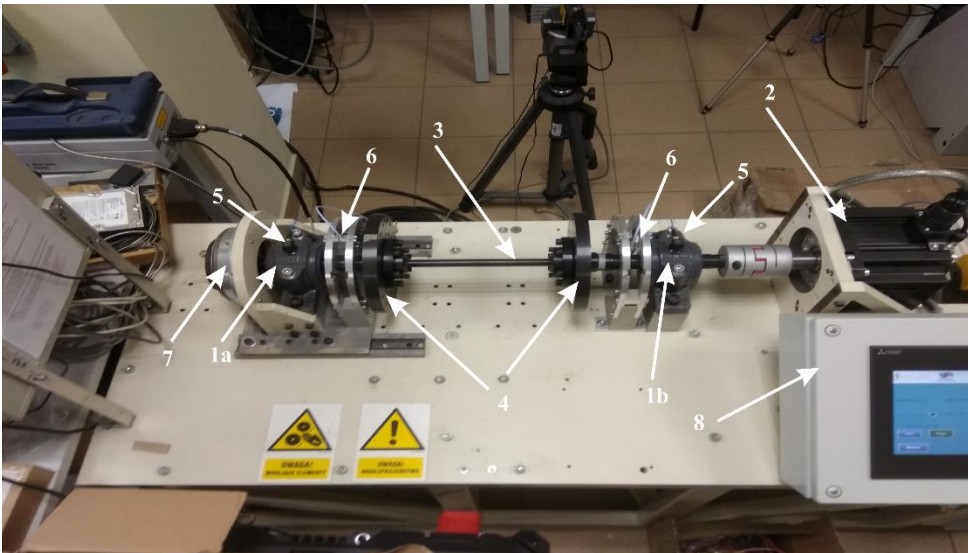

**Figure 2.** Measurement test rig: 1—ball bearings (a—on the brake side, b—on the drive side); 2—motor; 3—shaft; 4—mass disc; 5—accelerometers; 6—eddy current sensors; 7—electromagnetic brake; 8—control panel.

**Table 2.** Experimental test stand parameters.

| Parameter Description | Value [Unit] |
| --- | --- |
| Bearing type | SKF 1207K EKTN9 |
| Rotational speed | max 2 000 [rpm] |
| Shaft critical speed | 3200 [rpm] |
| Load torque | No external forces |
| Radial force | |
| Sampling time | 60 [s] |
| Sampling rate | 8192 [Hz] |

The object of the tests was the SKF 1207K bearing (Figure 3) with the parameters listed in Table 3:

**Table 3.** Parameters of ball bearing SKF 1207K EKTN9 [90].

| Parameter | Value [Unit] |
|---|---|
| Manufacturer's model | 1207 EKTN9 |
| Outer diameter | 72 [mm] |
| Inner diameter | 35 [mm] |
| Outer ring periphery diameter | 60.9 [mm] |
| Inner ring periphery diameter | 47 [mm] |
| Width | 17 [mm] |
| Weight | 0.32 [kg] |

The bearings were tested in five configurations: undamaged, with cut outer and inner race, with the cut ball at a depth of 1, 2, and 3 mm and width 0.3 mm. The registration of the diagnostic signals was implemented by eddy current sensors, positioned radially to the rotating shaft in the vertical Y and horizontal X axes (two sensors at each bearing).

The tested bearing is shown in Figure 3a and the types of its damage are presented in Figure 3b–f.

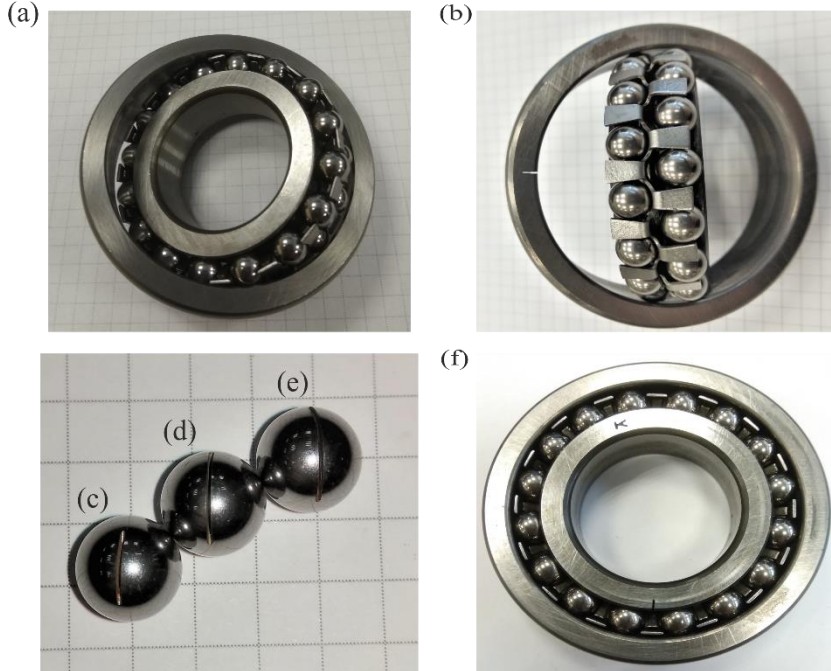

**Figure 3.** Ball bearing SKF 1207K: (**a**) undamaged; (**b**) with cut outer race; with the cut ball at the depthof: (**c**) 1 mm; (**d**) 2 mm; (**e**) 3 mm; (**f**) with cut inner race.

## 4. Experimental Results

To better show the type and influence of the simulated bearing faults on the measured vibrations, the authors prepared an FFT analysis of all damage signals for 900 rpm.

The frequencies of bearing 1207K, calculated for speed 900 rpm, are shown in Table 4:

**Table 4.** Frequencies of ball bearing SKF 1207K EKTN9 [90].

| Parameter | Value [Unit] |
|---|---|
| Rotational frequencies: | |
| Inner ring | 15 [Hz] |
| Outer ring | 0 [Hz] |
| Rolling element set and cage | 6.285 [Hz] |
| The rolling element about its axis | 44.581 [Hz] |
| Frequencies of over-rolling: | |
| A point on the inner ring | 130.72 [Hz] |
| A point on the outer ring | 94.28 [Hz] |
| Rolling element | 89.162 [Hz] |

Based on the expected frequencies (Table 4) and the obtained frequency characteristics (Figures 4 and 5), the bearing faults are visible. However, the damages are not significant in the frequency characteristics, this is due to the narrow cut on the bearing elements. More significant bearing faults are visible in the amplitude change and new frequencies appearance [41]. However, the authors wanted to introduce minor damage.

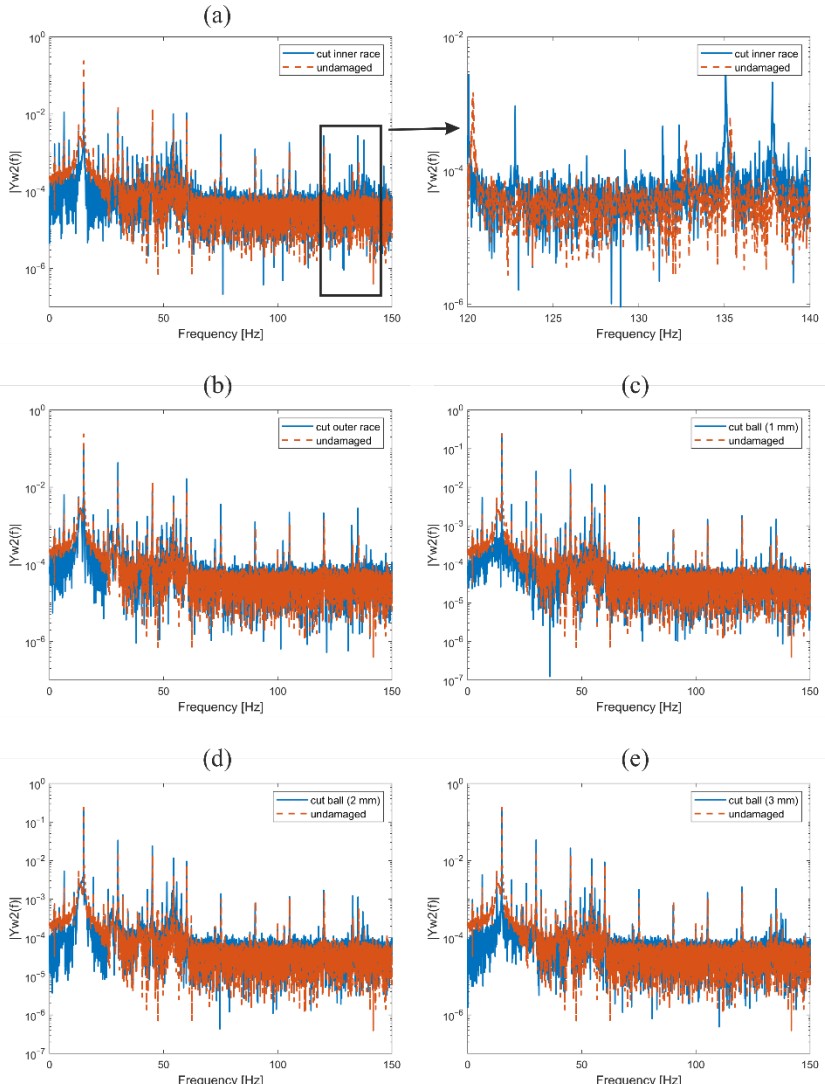

**Figure 4.** The FFT analysis of the recorded shaft displacement signal influenced by the vibration signal coming from bearing (drive side) SKF 1207K: (**a**) with cut inner race; (**b**) with cut outer race; with the cut ball at the depth of: (**c**) 1 mm; (**d**) 2 mm; (**e**) 3 mm.

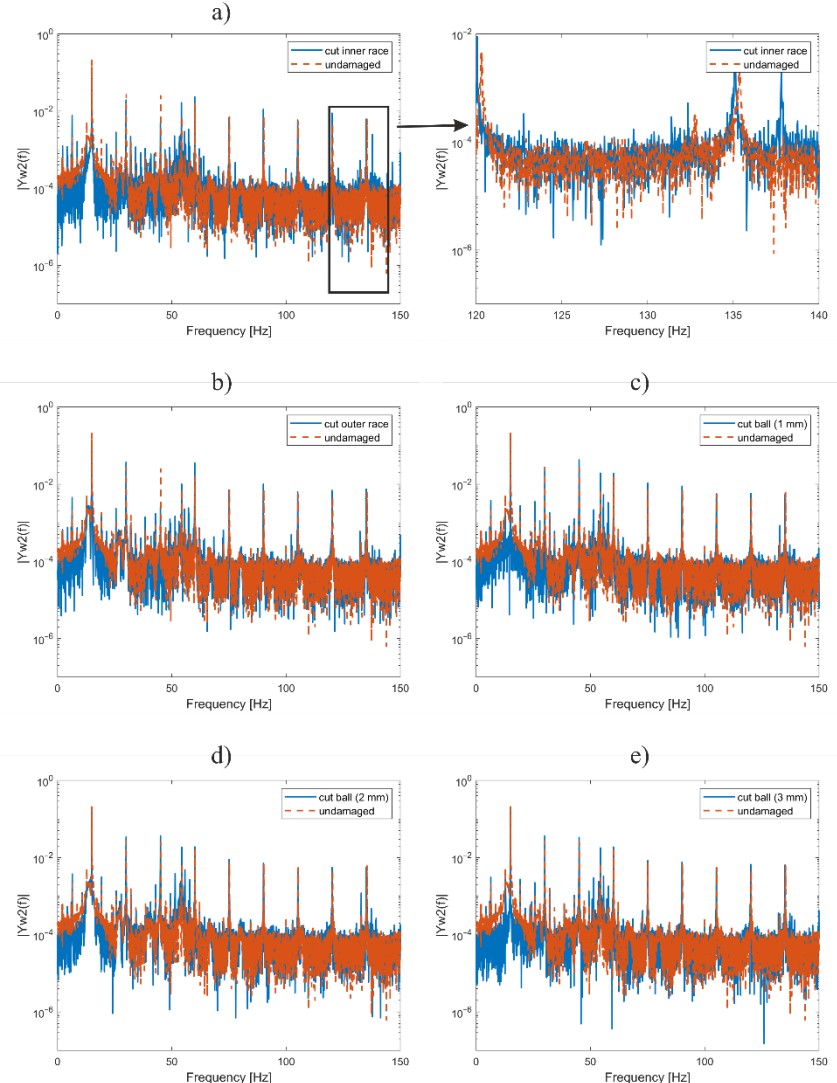

**Figure 5.** The FFT analysis of the recorded shaft displacement signal influenced by the vibration signal coming from bearing (brake side) SKF 1207K: (**a**) with cut inner race; (**b**) with cut outer race; with the cut ball at the depth of: (**c**) 1 mm; (**d**) 2 mm; (**e**) 3 mm.

Upon determining the frequency characteristic, the analysis was carried out using a squared amplitude gain method.

Examples of displacement signals of one shaft revolution obtained at the sampling frequency of 8192 Hz for each variant of bearing damage are shown in Figure 6.

The recorded experimental signals included the course of shaft vibrations during rotation for 60 s after the stabilization of the set rotational speed. In addition, measurements were carried out for rotational speeds from 100 every 100 to 2000 rpm and for each damage variant.

The shaft vibration signal from each sensor, containing many successive shaft revolutions, was divided into single revolutions. The analysis of the results was presented for the rotational speed of 900 rpm, and the signals were recorded for eddy current sensors located radially to the shaft in the vertical axis.

For the exemplary speed of 900 rpm, 900 cycles were obtained, corresponding to the number of shaft revolutions per minute. In each analysis, the first and last cycles were discarded, assuming they could represent an incomplete shaft rotation. For each of the cycles, the procedure was performed following the description of the model (Section 2). First, the Hanning window was used, and then the signal autocorrelation was determined.

The autocorrelation result was approximated by a polynomial, where the coefficient of determination R2 measured the level of the polynomial fit. Following some preliminary calculations, the order of approximating polynomials of the auto-correlation function was chosen as $n = 5$, which gives the coefficient of determination $R^2 > 99.97\%$. Adoption of the lower orders of polynomials leads to lower values of the coefficients of determination (i.e., less accurate approximations). Higher orders do not noticeably improve the approximation accuracy (because it is already close to 1), but they significantly extend the calculation time and the number of coefficients to analyze.

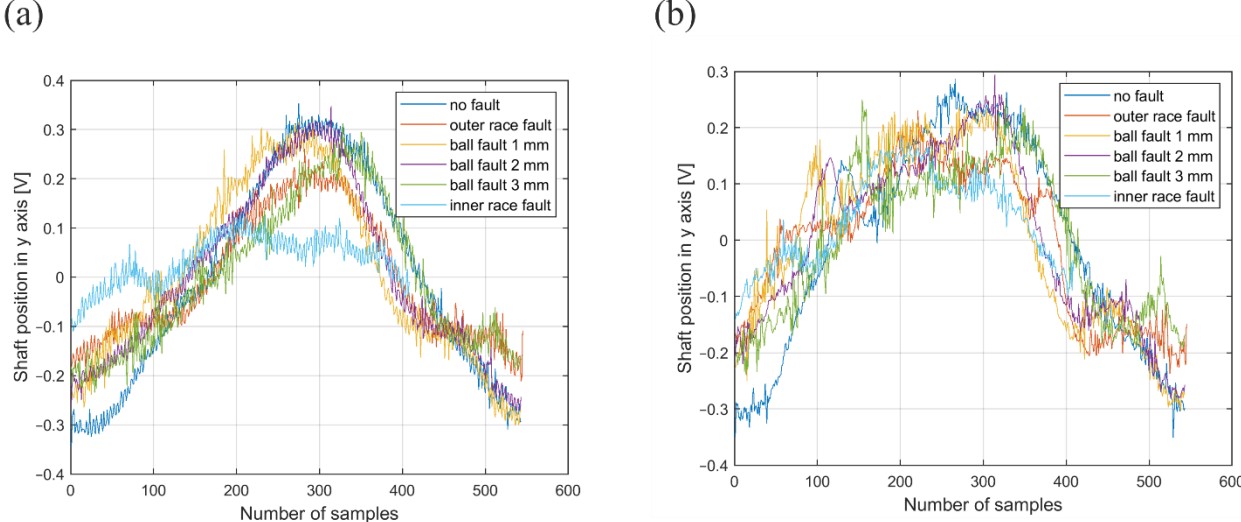

**Figure 6.** The recorded shaft displacement signal influenced by the vibration signal coming from the bearing: (**a**) from the brake side; (**b**) from the drive side.

Examples of auto-correlation functions and their fifth-order polynomial approximations in two observation zones of the selected cycle of the rotor displacement signal (from Figure 1) are shown in Figure 7.

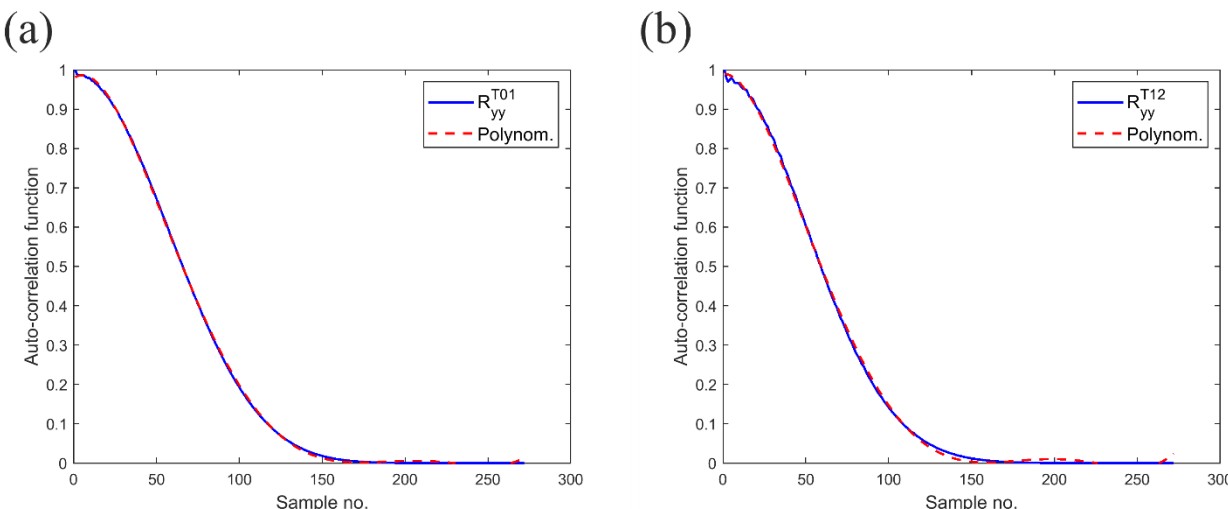

**Figure 7.** Auto-correlation signal for the rotor supported on the SKF 1207K bearing (with the ball cut on a depth of 1 mm) with the matching curve: (**a**) during the observation period $T_{01}$; (**b**) during the observation period $T_{12}$.

The damage map of the bearing technical condition consists of the 30 cycles, randomly selected (which is a statistical sample) from all (for a speed of 900 rpm, 900 cycles) recorded cycles, for a given speed. It is expected that the damage maps obtained for each draw will

give the same picture of the technical condition of the bearing (which will confirm the method's effectiveness).

First, a damage map was created for the undamaged bearing. The diagnostic thresholds set for it were the reference thresholds used for damaged bearings. The damage map contains 30 rows marked with numbers of randomly selected cycles and 12 columns, which consist of coefficients of 5th-order polynomials (six coefficients each) for the numerator and denominator of the model. Comparing the model results for different damage variants is possible, using only the same degree of the matching polynomial. The numbers of the damaged bearings cycles were the same as for the previously randomly selected cycles of the undamaged bearing.

The technical condition of the bearing is represented by 12 coefficients: six in the first observation period ($T_{01}$) and six in the second observation period ($T_{12}$). These zones are visible as the left and right parts of the technical condition damage maps (Figures 8a–f and 9a–f), separated by a thick black line.

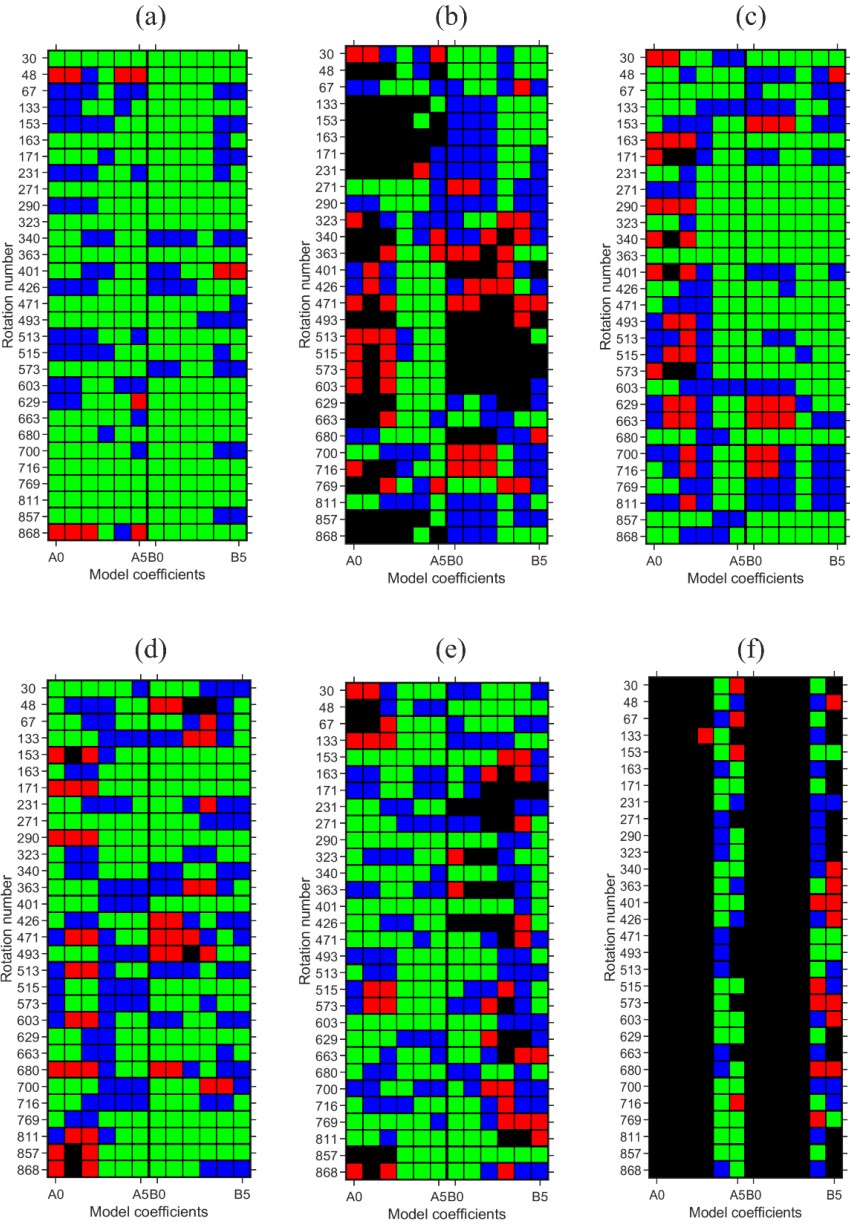

**Figure 8.** The damage map of bearing (drive side) SKF 1207K: (**a**) undamaged; (**b**) with a damaged outer race; with the cut ball at the depth of: (**c**) 1 mm; (**d**) 2 mm; (**e**) 3 mm; (**f**) with cut inner race.

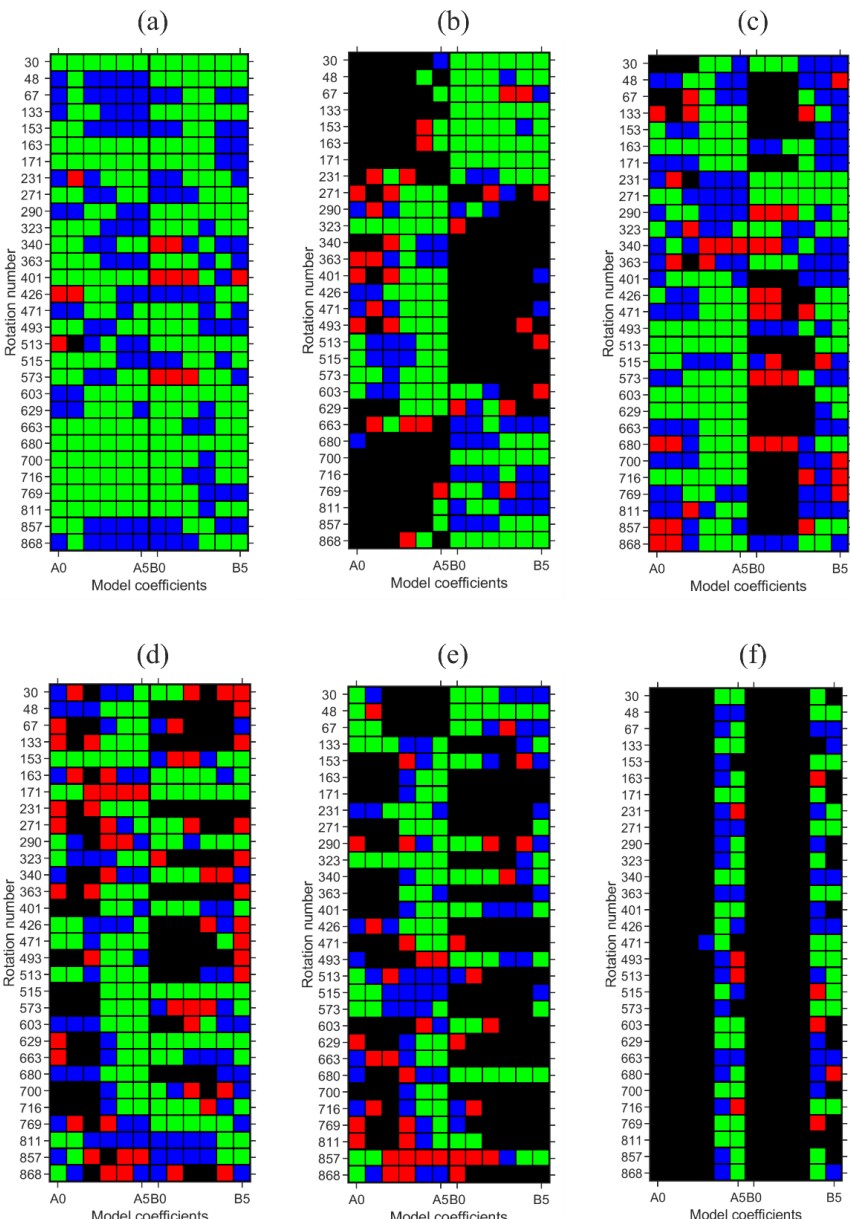

**Figure 9.** The damage map of bearing (brake side) SKF 1207K: (**a**) undamaged; (**b**) with a damaged outer race; with the cut ball at the depth of: (**c**) 1 mm; (**d**) 2 mm; (**e**) 3 mm; (**f**) with cut inner race.

Based on Figures 8 and 9, it can be seen that the technical condition of each bearing is clearly and unambiguously indicated on the damage maps obtained. Dominant *red* and *black* fields are characteristic of severely damaged bearings. Moreover, the undamaged bearing is dominated by *blue* and *green*. Since damage thresholds are based on statistics, slight variations between the colors of a given damage map are possible. Therefore, it is important to analyze the map globally for a reasonable number of rotation cycles (map rows).

Damage maps can present the technical condition of all analyzed bearings in a concise, graphical form. Based on the generated maps (Figures 8 and 9), it is possible to present a quantitative form of the technical condition. Collective graphs of quantitative analysis (Figure 10) will facilitate the comparison of damage maps in terms of the increase in the number of *black* and *red* fields and the decrease in *green* in relation to the increasing damage.

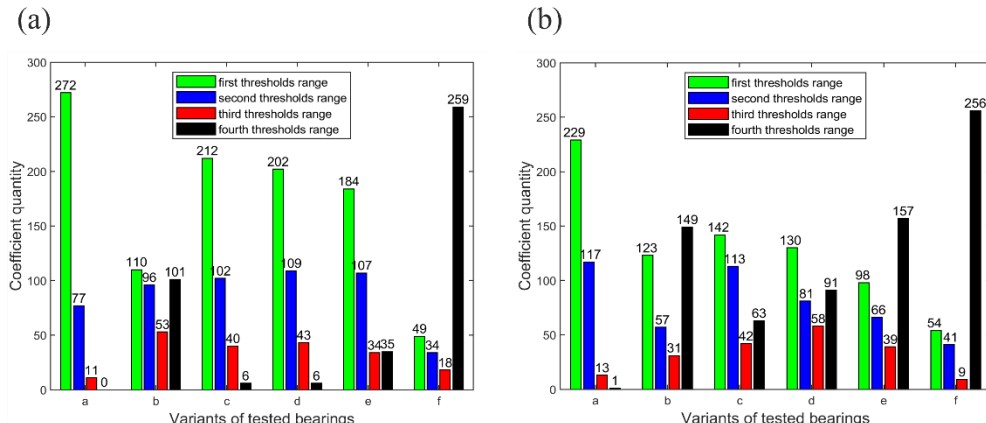

**Figure 10.** Collective characteristics of the damage maps of the SKF 1207K bearing from the side of: (**a**) the drive (**b**) the brake for the shaft speed of 900 rpm.

Figure 10 presents a quantitative summary of all tested bearing variants at the rotor speed of 900 rpm (Figure 10a-bearing closer to the brake, 10b-bearing closer to the drive). Bearing variants are marked on the horizontal axis (a–undamaged bearing; b–with a damaged outer race; with the cut ball at the depth of: c–1 mm; d–2 mm; e–3 mm; f–with a damaged inner race), and on the vertical axis the number occurrences of a given color. It can be seen in both types of figures that the undamaged variant significantly dominates over the others in the case of the threshold not exceeding $\mu \pm \sigma$ (*green*), while the values of the thresholds indicating damage have the lowest. It can also be noticed that in the case of bearing balls with cuts 1, 2 and 3 mm deep, the threshold $\mu \pm \sigma$ decreases (*green*) while the threshold $\mu \pm 2\sigma$ increases (*black*). High values of *black* and *red* colors are visible for the variant of the bearing's outer race damage, as a result of which, during the rotation of the shaft, all of the balls in the bearing hit the discontinuity of the race, generating vibrations. The highest values of *black* color can be seen for the bearing with the cut inner race. In this case, balls more frequently hit the cut point on the inner race than in the outer race.

Summarizing the results presented in the collective characteristics (Figure 10), the number of occurrences of the *black*, *red*, *blue*, and *green* color fields is collected in Table 5. As can be seen from the damage maps, where the damage to the ball was 1 mm deep (Figure 9c) and 2 mm deep (Figure 9d), it is difficult to see an increase in the number of parameters. Therefore, a good complement to the damage maps of the technical condition is their quantitative analysis, from which it is already clear that there was an increase in the number of *red* and *black* fields between the indicated damage.

**Table 5.** Number of color fields in subsequent damage maps presented in Figures 8 and 9.

| Selection | Bearing No 1 | | | | Bearing No 2 | | | |
|---|---|---|---|---|---|---|---|---|
| | *black* | *red* | *blue* | *green* | *black* | *red* | *blue* | *green* |
| Undamaged bearing | 0 | 11 | 77 | 272 | 1 | 13 | 117 | 229 |
| Damaged outer race | 101 | 53 | 96 | 110 | 149 | 31 | 57 | 123 |
| Cut ball at a depth of 1 mm | 6 | 40 | 102 | 212 | 63 | 42 | 113 | 142 |
| Cut ball at a depth of 2 mm | 6 | 43 | 109 | 202 | 91 | 56 | 81 | 130 |
| Cut ball at a depth of 3 mm | 35 | 34 | 107 | 184 | 157 | 39 | 66 | 96 |
| Damaged inner race | 259 | 18 | 34 | 49 | 256 | 9 | 41 | 54 |

The presented results reflect the technical condition of the bearing. In its current form, the method cannot determine the type of bearing damage based on the damage map, e.g., damage to the bearing ball to a depth of 1 or 3 mm. However, it is very effective for detecting when a bearing goes from a usefulness condition to failure (damage).

## 5. Summary

The parametric method of bearing failure detection presented in the article is based on auto-correlation and power spectral density functions. The signal is analyzed in two separate periods. When the space between the analyzed intervals is close to each other, the influence of external disturbances is eliminated. Therefore, each change in the parameters of the diagnostic model should be interpreted as a change in the machine's technical condition.

Markings of bearing damage are decipherable and presented in the form of characteristic color maps, where dominant *green* and *blue* indicate an undamaged bearing or very low wear, *red*—increased wear, or *black*—bearing damage. The exact location of the colors on the damage maps may vary as the diagnostic thresholds are determined by the mean and standard deviation, i.e., statistical parameters. Therefore, it is not the exact locations that are important but the predominance of a particular color. It should also be noted that the described parametric method should not be used on small datasets because it requires the determination of diagnostic thresholds based on statistical operations.

The experimental results confirmed that the method can reliably detect bearing damage. Furthermore, the method is simple and uses only measured vibration data (signal from one sensor is sufficient). Therefore, preliminary preparation of the rotor for testing is not required. Furthermore, displacement is measured on the shaft, so the bearing could be inaccessible or difficult to reach. In addition, the machine can be continuously monitored online. This conclusion allows for the future practical implementation of the method.

At the current stage of development, the method can only warn about the detection of bearing damage (the method cannot indicate the type of damage). Therefore, informing the maintenance staff about the need to carefully look at the bearing (when *blue* and *red* colors dominates) or to replace it (when the damage map is mostly *red* and *black*) becomes a possibility.

The next stages of the method developed in diagnosing bearings will be continued in subsequent articles.

**Author Contributions:** R.G.: Conceptualization, Investigation, Methodology, Project administration, Validation, Visualization, Writing—Original Draft, Funding acquisition, Resources, Supervision. B.B.: Data curation, Investigation, Resources, Software, Validation, Visualization, Writing—Reviewing and Editing, Funding acquisition. J.E.M.: Reviewing and Editing, Visualization, Funding acquisition. All authors have read and agreed to the published version of the manuscript.

**Funding:** The presented work was supported by the Ministry of Education and Science in Poland (research project No. WZ/WM-IIM/2/2022), by the Polish National Agency for Academic Exchange as part of the Academic International Partnerships (PPI/APM/2018/1/00033/U/001 project) and Erasmus+ Programme fund (partnership agreement 2017–3475/001–001).

**Institutional Review Board Statement:** Not applicable.

**Informed Consent Statement:** Not applicable.

**Data Availability Statement:** Not applicable.

**Conflicts of Interest:** The authors declare that they have no conflict of interest.

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
