# Peer review of "Bearing Fault Diagnostics Based on the Square of the Amplitude Gains Method"

_applsci, doi:10.3390/app13042160_

Round 1
Reviewer 1 Report
Abstract should be written in proper manner. I suggest to remove the citations from there, and make it in proper way.
The literature review is not sufficient. Need proper references, and justification.
Do the authors have any plan to incorporate this study with any publicly available dataset? If possible I suggest to do that. If not, then discuss about the possiblities.
Author Response
In attachment is a file with responses to the reviewer's comments

Reviewer 2 Report
1)In the introduction, the review of the research progress of the square of the amplitude gains method should be enhanced.
2)In line 109, What value of n is finally determined?
3)In line 114,x(t) is approximately subtracted later,but,the expressions for y(t) need to be added.
4)In line 151, It is recommended that there be a quantitative grading scale for the degree of failure, rather than a qualitative analysis.
5)Determining the type of failure based on damage map is a more meaningful study.
Author Response

(The authors gave the same response as above.)

Reviewer 3 Report
Manuscript entitled “Bearing fault diagnostics based on the square of the amplitude gains” is a very weak paper with the contributions being unclear.
This paper is not well arranged. Except English writing, there are many language expressions and formatting errors in the manuscript. So I recommend to you that this manuscript can not be accepted. The following are the questions and some mistakes in this manuscript:
1. This paper proposed a new method for bearing fault diagnosis, but the proposed method should be compared with other methods of bearing fault diagnosis and it is necessary to explain why the proposed method is better or different from other methods. (INTRODUCTION - literature data, RESULTS AND DISCUSSION - discussion)
2. It has to be outlined what is the benefit of the proposed method. (ABSTRACT, RESULTS AND DISCUSSION, CONCLUSIONS)
3.There is no experimental comparison of the proposed method with previously known work in the experimental section, so it is impossible to judge the superiority of the proposed method.
4. In page 1, the author mentioned that: " Accurate detection and removal of early stages of bearing damage contribute…”, I don't think it's appropriate to use the word "removal" here. Please read your paper carefully and ensure that your material is properly prepared and formatted before submitting your manuscript.
Author Response

(The authors gave the same response as above.)

Round 2
Reviewer 1 Report
Comments:
1. Please make sure the abstract is in one paragraph.
2. In the introduction, please include the reference for Stockwell Transform (time-frequency methods). You can cite this https://doi.org/10.3390/app8122357
Thank you.
Author Response
All answers are in the attachment

Reviewer 2 Report
No more comments.
Author Response
All answers are in the attachment
